# The Drivers, Mechanisms, and Consequences of Genome Instability in HPV-Driven Cancers

**DOI:** 10.3390/cancers14194623

**Published:** 2022-09-23

**Authors:** Vanessa L. Porter, Marco A. Marra

**Affiliations:** 1Canada’s Michael Smith Genome Sciences Centre, BC Cancer, Vancouver, BC V5Z 4S6, Canada; 2Department of Medical Genetics, University of British Columbia, Vancouver, BC V6T 1Z4, Canada

**Keywords:** HPV, virus, cancer, DNA damage, genome instability, viral integration, mutation, cervical cancer, head and neck cancer, DNA repair

## Abstract

**Simple Summary:**

Cells infected with high-risk human papillomaviruses (HPV) can accumulate DNA damage and eventually transform into HPV-driven cancers. Genome instability, or the progressive accumulation of DNA alterations (e.g., mutations), in HPV-infected cells is directly induced by the HPV genes and indirectly promoted by HPV infection through the consequences of chronic infection maintenance, increased cell growth, and accumulation of damaging mutations in genes that themselves affect genome instability. While the HPV genome typically exists as a separate entity within cells, genome instability increases the chances of HPV integrating within the host (human) genome, which is common in HPV-induced cancers. The DNA regions surrounding HPV integrations are unstable and can undergo complex alterations that affect both human and HPV genes. This review discusses HPV-dependent and -independent drivers and mechanisms of genome instability in HPV-driven cancers, both globally and around sites of HPV integration, and describes the changes induced in the tumour genome.

**Abstract:**

Human papillomavirus (HPV) is the causative driver of cervical cancer and a contributing risk factor of head and neck cancer and several anogenital cancers. HPV’s ability to induce genome instability contributes to its oncogenicity. HPV genes can induce genome instability in several ways, including modulating the cell cycle to favour proliferation, interacting with DNA damage repair pathways to bring high-fidelity repair pathways to viral episomes and away from the host genome, inducing DNA-damaging oxidative stress, and altering the length of telomeres. In addition, the presence of a chronic viral infection can lead to immune responses that also cause genome instability of the infected tissue. The HPV genome can become integrated into the host genome during HPV-induced tumorigenesis. Viral integration requires double-stranded breaks on the DNA; therefore, regions around the integration event are prone to structural alterations and themselves are targets of genome instability. In this review, we present the mechanisms by which HPV-dependent and -independent genome instability is initiated and maintained in HPV-driven cancers, both across the genome and at regions of HPV integration.

## 1. Introduction

Loss of genome stability, either through mutation of a critical tumour suppressor or chronic carcinogenic exposures, is a central hallmark of cancer [1,2,3]. Viral infection has been linked to 9.9% of all cancers worldwide and can contribute to oncogenic transformation by increasing cellular proliferation and dysregulating host cell genomes, often through the integration of the viral genome into the host cell’s DNA [4,5,6]. Human papillomavirus (HPV) accounts for 4.5% of all cancers and approximately half of the viral cancers [7]. A few types of HPV, most commonly HPV16 and HPV18, are the predominant causes of cervical cancer and a contributing risk factor for several anogenital cancers and head and neck squamous cell carcinoma (HNSCC) [7,8]. Vaccination against the most prevalent oncogenic HPV types has reduced cervical cancer incidence in young females [9], but in males where the vaccination campaign has begun more recently and in males and females from low- and middle-income countries where vaccine availability is an issue, HPV-associated cancers persist [10,11,12]. Given the endemic nature of HPV and its association with cancers in populations around the world, it is of critical importance to understand how HPV infection can disrupt genome structure and function to contribute to malignant progression. Insights gleaned from such work may be applicable to other less-well studied oncogenic viruses.

Genomic instability refers to an increased prevalence of alterations that get passed down to daughter cells upon cell division [1]. These alterations can range from single nucleotide variants (SNVs) to large-scale structural variants (SVs) involving multiple chromosomes. The transmission of SVs and SNVs can occur at an increased frequency upon dysregulation of different DNA repair pathways [13,14]. Additionally, chronic exposure to exogenous mutagens, such as smoking, sun exposure, and chemotherapeutics, can leave a characteristic mutation signature on tumour DNA [15,16,17]. These patterns can contribute mechanistic insights into drivers of genomic instability in the tumour and, in some cases, such information may inform patient treatment and prognosis [15,18]. In 80% of cervical cancers and between 25–70% of head and neck cancers depending on the study, portions of the HPV genome are integrated into the cancer cell genome, and these integrated regions undergo transformative rearrangements as a consequence [19,20,21,22]. In this review, we explore the drivers, mechanisms, and consequences of genomic instability in HPV-driven cancers, both across the cancer genome and at the sites of HPV integration.

## 2. The HPV Life Cycle and Oncogenesis

### 2.1. Expression of HPV Genes through the Viral Life Cycle

Over 400 types of HPV have been identified, all of which are tropic for squamous cells on different parts of the body, though the tropisms of individual HPV types are remarkably specific [23]. The 14 HPV types that have been designated as carcinogens by the World Health Organization (WHO) belong in the alpha-papillomavirus family and infect squamous cells in mucosal tissues, including (but not limited to) the transformation zone between the ectocervix and the endocervix, and the tonsillar crypts of the oropharynx [23,24,25,26]. Tissue micro-abrasions within these regions allow oncogenic HPVs to infect basal stem cells during sexual activity [27]. Once introduced, the expression of HPV genes (Figure 1a) as well as the viral life cycle becomes tied to the differentiation process of the epithelium [28]. In the undifferentiated basal and parabasal layers, viral genome replication is initiated by the HPV proteins E1 and E2, with the DNA binding protein E2 recruiting the helicase E1 to the viral replication origin while also tethering viral episomes to host chromatin for a symmetrical partition of replicated episomes between the two daughter cells [29,30,31]. In the mid-epithelial layers, high expression of the *E6* and *E7* oncogenes cause differentiated keratinocytes to re-enter the cell cycle, resulting in an induction of proliferation [32,33]. Once the HPV genome is amplified to about 100–200 episome copies per cell, E2 negatively regulates the early genes, including the oncogenes *E6* and *E7*, by binding the viral non-coding long control region (LCR) upstream of the early promoter, thus preventing the binding of transcription factors and the transcription initiation complex to this regulatory region [34]. Terminal differentiation of the keratinocytes is triggered when E2 binds the early promoter to down-regulate the early genes and the viral transcript changes its splicing to favour the late-differentiation genes *E4*, *L1*, and *L2* [35]. The late genes, *L1* and *L2*, make up the capsid proteins of the virion that are needed to encapsulate the viral DNA upon viral release and reinfection [36], and the packaged virus is released upon cell death [37].

### 2.2. Transformation of Infected Cells into Cancer

Infection does not always lead to cancer. Most asymptomatic or low-grade HPV infections are cleared within two years by the immune system or through sloughing of the cervical epithelium [38]. However, if the immune system is weak or unable to clear the infection, then infection can persist and the infected cells can slowly accumulate DNA damage. Active infection by oncogenic HPV types usually leads to increased proliferation of the mid-epithelial layers, which can present as precancerous lesions such as cervical intraepithelial neoplasia (CIN) [39] (Figure 2). It is suspected that precancerous lesions also exist in the oropharynx prior to cancer, but there are no routine screening procedures in place to catch asymptomatic infections in the head and neck [40]. Cervical screening, however, allows early detection of CIN so patients can receive intervention and treatment before these low-grade lesions transform into malignant cancer [41]. If these lesions are left unchecked, more mutations can be acquired that increase proliferative potential, halt keratinocyte differentiation, or lead to the evasion of apoptosis [42]. Integration of HPV into the host genome frequently occurs during the transformation of infected cells into cancer [4]. This process usually truncates or removes *E2* from the integrated form of HPV, resulting in improper regulation of expression of *E6* and *E7* (Figure 1b) [43,44]. After losing the ability to properly regulate the HPV oncogenes, keratinocytes carrying integrated HPV get trapped in the proliferative amplification step of the viral life cycle and can no longer exit the cell cycle [42,45]. Once the abnormal cells have broken through the basal cells in the epithelium, the lesion is then graded as an invasive cervical carcinoma (ICC) and becomes increasingly difficult to treat [46]. The transformation of HPV infection into cancer is not advantageous for the virus since it effectively eliminates its ability to reinfect new hosts, therefore terminating its life cycle [39]. HPV-induced oncogenesis is thus the unfortunate result of the accumulation of genetic mutations that are accelerated by the presence of a chronic infection and de novo introduction of viral oncogenes.

## 3. Direct Effects of the HPV Genes on Genome Instability

### 3.1. Cell Cycle Dysregulation Allows an Accumulation of Unchecked DNA Damage

HPV-driven cancers require high expression of the HPV oncogenes even after malignant transformation [6]. All oncogenic HPV types include the oncogenes *E6* and *E7*, which target p53 and Rb family proteins for degradation, respectively [47,48]. The functional loss of these prominent tumour suppressors leaves the genome vulnerable to the accumulation of mutations, which eventually can lead to cancer. Many cell culture studies over the past 30 years have shown that expression of *E6* and *E7* from either HPV16 or HPV18 is sufficient to immortalize keratinocytes [49,50,51,52]. When each gene was tested alone, only *E7* expression and not *E6* had transformative potential [53]. However, *E7*-expressing human uroepithelial cells (HUC) remained relatively genotypically stable while *E6*-expressing HUC cells accumulated chromosomal aberrations with increasing passages [54]. Another early study showed that *E6*-expressing cells that were exposed to UV radiation were unable to recruit the necessary repair pathways to DNA damage in the genome and therefore had reduced viability after the exposure [55]. These results showed that both *E6* and *E7* expression are important during tumorigenesis: *E7* for establishing cell immortalization and *E6* for promoting genomic instability that can lead to malignant progression. Although both E6 and E7 are multifunctional with a large number of cellular targets, the tumorigenic effects of these proteins are thought to be largely related to their effects on cell cycle dysregulation (Figure 3).

The main human interacting partner of E6 is the E3 ubiquitin ligase E6 Associated Protein (E6AP) (Figure 3) [56]. The most notable and well-studied effect of this interaction is the assembly of the E6/E6AP/p53 protein complex [57]. The E6 protein binds to the leucine-rich LxxLL motif of E6AP and then undergoes a conformational change that allows it to bind p53 on a site distal to its DNA- and protein-interacting domains, allowing the complex to form whether or not p53 is bound to a target [57]. The interaction of E6/p53 with E6AP leads to the ubiquitination of all members in the E6/E6AP/p53 complex and their subsequent targeted proteasomal degradation [58,59,60]. Any remaining p53 that was not degraded remains inhibited by E6 from activating its targets [61]. HPV-infected cells are therefore p53-deficient at the post-translational level; thus, somatic mutations in *TP53* do not lead to a selective advantage in HPV-associated tumours [62]. The p53 protein is known as the major tumour suppressor of the cell, involved in cell cycle arrest, DNA repair, senescence, apoptosis, and an increasing number of other important pathways [63]. The absence of the p53 DNA damage checkpoint in the proliferating HPV-infected cells allows the cells to proceed into S-phase with unchecked and unrepaired DNA damage.

Meanwhile, E7 drives cell cycle progression through its direct interaction with the Retinoblastoma protein family (pRb) members (Figure 3) [33]. The binding of E7 to pRb prevents its binding to the E2F family of transcription factors, allowing the transcription of S-phase genes [64]. Both low-risk and high-risk HPV types require E7 to uncouple keratinocyte differentiation from its requirement to exit the cell cycle through its inhibition of pRb, allowing the mid-epithelial layers to proliferate [65]. However, high-risk HPV types contain *E7* genes with protein products that have a higher binding affinity to a wider range of Rb-related proteins (e.g., p107 and p130) than non-oncogenic HPV types, resulting in a more robust induction of proliferation [64]. Independent from pRb, E7 also promotes cell cycle progression by activating E2F1 and inactivating the negative-feedback counterpart of E2F transcription, E2F6 (Figure 3) [66,67]. E7 also promotes G1/S transition through interactions with cyclins and cyclin-dependent kinases (CDKs). The cyclins are cell cycle regulators that bind to CDKs to drive or inhibit their activation of downstream target proteins that control processes important in specific cell cycle phases [68]. E7 drives transcriptional and post-translational activation of both cyclin A and cyclin E (Figure 3), enabling entry into S-phase without the induction of cyclin D1, which is usually required for G1/S progression [69,70,71,72]. Both CDK1 and CDK2 are upregulated in E7-expressing cells, but B-Myb-dependent upregulation of CDK1 was found to be specifically important in bypassing the G1 DNA damage checkpoint (Figure 3) [73]. Abrogation of the G1 checkpoint has been found to be mediated by RCC1 through CDK1, and overexpression of CDK1 can rescue the progression to S-phase in E7-expressing cells that have stalled cell cycle progression due to knockdown of RCC1 [74,75]. G1/S progression, in the absence of the DNA damage G1 checkpoint, results in the accumulation of chromosome abnormalities.

The HPV protein E5 also has functions in modulating the cell cycle, though its role and importance in HPV-induced tumorigenesis is not as defined as E6 and E7. Unlike E6/E7, E5 alone cannot immortalize keratinocytes, but it can potentiate and enhance the effects of E6/E7-induced immortalization [76]. Specifically, E5 was found to synergize with E7 to promote premature, unchecked DNA synthesis, which in turn led to increased cell proliferation and more severe cervical dysplasia in a transgenic mouse model [77]. One known carcinogenic mediator of E5 is the epithelial growth factor receptor (EGFR) signalling pathway [78]. E5 has been shown to increase the number of membrane-bound EGFR through inhibiting the endosomal degradation of EGFR and also by the alteration of protein trafficking in the cell [79,80]. In addition, E5 has been found to mediate the activation of the pro-proliferation EGF, PI3K-AKT, and MAPK signalling cascades through EGFR-dependent and -independent processes [81,82,83,84]. The *E5* gene is not consistently expressed in HPV-driven cancers, but its presence may aid in perpetuating uncontrolled cell cycling in tumours [85,86].

Importantly, the expression of *E6* and *E7* is also associated with mitotic defects, such as centrosome amplification and lagging chromosomes that result from defects in cell cycle checkpoints [87,88]. Polyploidy is also common among HPV-driven cancers. For example, Fan and Chen [73] demonstrated that genome re-replication can occur while HPV-infected cells are arrested at the G2 spindle assembly checkpoint, possibly through E7-induced expression of the DNA replication initiation factor CDT1. Thus, HPV-mediated dysregulation of the cell cycle drives genome instability by allowing both genetic mutations and chromosomal abnormalities to be passed down to daughter cells.

### 3.2. Interactions with the Host DNA Damage Response

DNA damage and replication stress can occur normally during DNA replication; however, human cells have high-fidelity DNA damage response (DDR) mechanisms in place to ensure that damage is repaired. The ATM-mediated DNA damage pathway is generally activated after double-stranded breaks [89]. The first responders to double-stranded breaks are the members of the MRN complex: MRE11, RAD50, and NBS1 [90]. The endonuclease MRE11 cleaves off nucleotides at the site of the break, RAD50 stabilises the broken DNA, and NBS1 recruits ATM [91]. Activated ATM then initiates a signalling cascade that recruits homologous recombination (HR) proteins such as BRCA1 and BRCA2 to facilitate high-fidelity repair using the intact homologous chromosome as a template [89]. Conversely, the ATR-mediated DDR pathway is responsible for resolving single-stranded breaks that occur during replication stress and fork stalling [92]. Single-stranded DNA is a normal intermediate in transcription and replication, but it can persist if the replication fork is stalled for reasons such as nucleotide insufficiency, irregular DNA structures, loss of open chromatin, or clashing transcription and replication forks [92]. The persisting single-stranded DNA, which is coated by the protein RPA, recruits ATR and its interacting partner ATRIP, as well as the 9-1-1 complex composed of RAD9, RAD1, and HUS1 [93]. The 9-1-1 complex recruits the topoisomerase TOPBP1 while activated ATR phosphorylates CHK1, and these proteins coordinate with downstream processes to repair the damage and restart the stalled replication fork [93].

Although p53 is not able to activate the DDR in HPV-infected cells, the ATM and ATR DNA damage sensing pathways are constitutively activated by the virus during HPV infection and are recruited to HPV episomes, resulting in high-fidelity viral replication and amplification [94]. Additionally, the expression of DDR proteins significantly increases as CIN progresses [95]. Viral episomes are transcribed and replicated at high levels, leading to replication stress that needs to be relieved by ATR-mediated repair [96]. DNA repair genes, including downstream mediators of the ATR pathways, were found to be specifically upregulated in HPV-positive HNSCCs when compared to their HPV-negative counterparts [97,98]. Studies have shown that as HPV-infected keratinocytes differentiate and the viral life cycle completes, the ATM pathway becomes upregulated and is necessary for productive episomal amplification [99,100]. This reliance is dependent on the stage of the viral life cycle: stable maintenance of episomes in undifferentiated keratinocytes can occur through low-level replication independently of ATM signalling, but differentiated keratinocytes in the productive stage of infection cannot form replication foci or amplify the HPV genome in the absence of the ATM response or HR effector proteins [99]. Since the DDR is essential for the HPV life cycle, several of the HPV proteins have direct and indirect interactions with DDR proteins. E7 is the main mediator for recruiting the DDR, which occurs through direct interactions with MRN complex members ABS1 and RAD51 (Figure 3) [100]. Additionally, E1 facilitates the recruitment of DDR proteins to the replication foci of amplifying episomes, while E2 can induce the ATM signalling response [29,101]. The E1/E2 initiation of episome replication through HR was found to be stabilised by the helicase WRN, specifically when WRN is deacetylated by SIRT1 [102]. However, *WRN* and *SIRT1* mRNA levels have an inverse relationship in CIN and cervical cancer; *WRN* levels reduce with increasing CIN grades while *SIRT1* increases [103]. Accordingly, the knockdown of WRN results in increased DNA damage in the HPV16+ cells [103]. In addition, E6 and E7 both cooperate to increase the protein levels of HR mediators by increasing gene expression and mediating protein stabilization [104,105]. E7 has also been shown to indirectly induce the DDR in response to endogenous and exogenous mutagens through an upregulation of *TP63* (Figure 3) [106]. Modulation and recruitment of the DDR, particularly homology-directed repair, is necessary for the continuation of the HPV life cycle [107].

Activation of the innate immune response can also trigger a DDR, which HPV can use to its advantage for viral genome replication [108]. As a first line of defence, Toll-like receptors (TLR) recognise pathogen-associated molecular patterns (PAMPs) for pathogen defence, but TLRs also can recognise damage-associated molecular patterns (DAMPs) created from DNA damaging events such as genotoxic stress and viral infection and subsequently activate the DDR pathways [109]. Similarly, the persistence of cytosolic DNA indicative of microbial infection or DNA damage can activate the STING pathway [110]. HPV-driven cancers have increased expression of these innate immunity pathways [111,112]. This upregulation can be owed in part to an increased immunogenicity of the tumour due to the viral presence, but direct interactions between HPV genes and TLRs and other immune effectors have also been reported [108,112]. HPV targeting of the innate immune response thus may aid in promoting DDR-mediated viral replication.

While HPV infection leads to a constitutively active DDR, this appears to come at the cost of host genome integrity. This is evident in HPV-driven cancer genomes, which generally show a high rate of HR deficiency and structural variants resulting from double-stranded breaks [113,114]. The viral genome in its episomal form, on the other hand, accumulates double stranded breaks less frequently than the host genome due to preferential recruitment of HR effector proteins to the viral episomes [115]. While the ATR pathway remains activated due to the replication stress generated from the virally-induced high replication rate [96], the recruitment of DDR proteins specifically to HPV foci can lead to host genome instability due to the insufficient availability of repair proteins. For example, Sitz et al. showed that E7 directly interacts with RNF168 (Figure 3), an E3 ubiquitin ligase that ubiquitinates the H2A histone tail on chromatin surrounding a double stranded break, which is important for recruiting downstream DDR effector proteins [116]. In HPV-infected cells, RNF168 and the downstream effectors are hijacked by the viral replication foci, which the authors hypothesize restricts DNA repair on the host genome [116]. Although HR pathway genes are upregulated in cervical cancer cells compared to healthy controls, Wallace et al. showed that *E6* and *E7* transduction reduces cells’ ability to repair double-stranded breaks on host DNA by about 50% [104]. The authors speculated that this reduction was due to the HR response being initiated during the G1 phase of the cell cycle when there is no homologous template available instead of during the G2/S phase following replication, and to RAD51 foci, which normally coat DNA breaks, being re-directed away from DNA breaks in the cell genome towards the viral episomes [104]. Altogether, the expression of high-risk HPV proteins, mainly E6 and E7, modulates the expression and localization of DDR genes and their products, respectively, to favour the HPV episomes, resulting in increased vulnerability of the host genome to double-stranded breaks.

### 3.3. Generation of Oxidative Stress

A byproduct of normal aerobic metabolism is the generation of highly reactive free radicals known collectively as reactive oxygen species (ROS). Cells attempt to balance these oxidative species with antioxidant enzymes and molecules to keep a balanced reduction-oxidation (redox) state [117]. The overproduction of ROS can occur when cells are in a state of stress, such as during infection, inflammation, chemical exposure, injury, and cancer [117]. The imbalance of oxidant to antioxidant molecules can cause oxidative and nitrative stress [118]. In addition to interfering with redox-sensitive proteins in the cell, unneutralized ROS can interact with DNA and leave oxidised guanine derivatives known as 8-Oxo-2’-deoxyguanosine (8-oxo-dG) [119], which can serve as a biomarker for increasing amounts of oxidative stress within a cell. The most common result of ROS damage to the genome are G→T transversions [120]. This can cause SNVs in the genome and also greatly affect the epigenome if CpG dinucleotides that are targeted by DNA methylation are mutated [121].

An unbalanced redox state has been shown to affect the integrity of the genomes of HPV-associated cancers. In cervical cancer, levels of 8-nitroguanine (resulting from reactive nitrogen species) and 8-oxodG were correlated with an increased cervical intraepithelial neoplasia (CIN) grade (grades 1–3) in HPV-infected cervical tissue [122,123]; the increasing oxidative and nitrative stress could either be due to the general presence of infection and cancer or a direct consequence of the HPV proteins. Cruz-Gregorio et al. showed that the early genes *E1*, *E2*, *E6*, and *E7* all have different individual effects on the redox state and that, in some cases, the combinatorial effect of multiple genes together can act synergistically or antagonistically [124]. The authors found that when *E6* alone was introduced in HPV-negative cervical cancer cells, ROS levels increased as well as DNA damage; however, transducing both *E6* and *E7* did not significantly alter the ROS levels [124]. *E1* and *E2*, when co-expressed, increased both ROS levels and γH2AX deposition [124]. The splice variant of E6 that is most highly expressed in cancers, E6*I, may be specifically responsible for ROS induction and the resultant DNA damage [125,126]. High-risk HPV types have *E6* and *E7* transcribed from the same promoter on a bicistronic transcript; the *E6* open reading frame also contains an intron that can be alternatively spliced to produce different *E6* splice forms [127,128]. Unspliced *E6* produces an E6 protein capable of its key oncogenic functions, while the *E6*I* splicing allows a more efficient translation of *E7* because it provides space between the *E6* stop codon and *E7* start codon [129,130]. Paget-Bailly et al. found that E6*I transcriptionally upregulates genes involved in NOX-dependent ROS production (*CCL2* and *RAC2*), while unspliced E6 downregulates these genes; as a result, unspliced E6 can offset E6*I-mediated ROS induction when co-transfected (Figure 3) [125]. Interestingly, the authors suggest that the *E6*I* mRNA may be responsible for these transcriptomic changes rather than its protein product, though the mechanism remains unknown [125]. NOX-induced oxidative stress and the resultant mutations were also observed in HPV-positive HNSCC cell lines as a consequence of *E6* and *E7* expression [131].

### 3.4. Alteration of Telomeres

Chromosome ends are protected from shortening after DNA replication by telomeric repeats, which naturally shorten with ageing [132]. The enzyme responsible for expanding telomere repeats, telomerase (TERT), is commonly subjected to mutation or dysregulation in cancer [133]. Telomere erosion and elongation can both have carcinogenic effects. Expanding the telomere repeats allows cells to replicate indefinitely, thus immortalizing them [134]. On the other hand, the erosion of telomeres can leave chromosomes vulnerable to chromosomal rearrangements, such as anaphase bridges and breakage-fusion-bridge cycles, since they may be interpreted as loose blunt ends by DDR machinery [135].

Telomere lengthening by HPV has generally been attributed to E6, while E7′s effect on telomeres has been shown to shorten and lengthen telomeres in different instances. One study found that low *TERT* expression and excessive telomere shortening in *E6*/*E7*-transfected cells correlated with an increased frequency of anaphase bridges a marker of unsuccessful chromosome segregation [136]. This effect was most prominent when *E7* alone was expressed and the normal phenotype could be rescued by the introduction of *TERT* [136]. However, contrasting evidence showed E7 restoring lost telomeres independently of TERT through induction of the alternative lengthening of the telomeres (ALT) pathway dependent on the protein FANCD2 (Figure 3) [137]. In this example, E7 stimulated the formation of ALT-associated promyelocytic leukemia (PML) bodies subnuclear structures that can predict telomere lengthening by ALT-dependent mechanisms and that these contained FANCD2 along with single-stranded DNA that co-localized with ATR-dependent repair mediators [137].

Conversely, E6 promotes telomere elongation through the activation of the *TERT* gene (Figure 3). Normally, *TERT* is not expressed or is expressed at low levels in keratinocytes; however, one study found that 64% of cervical cancers displayed high *TERT* expression [138]. In one study, when *E6* was introduced, the repressor complex USF1/USF2 that usually occupies the *TERT* promoter was replaced by c-MYC and *TERT* expression was induced [139]. However, others have suggested that *TERT* induction by E6 can occur independently of c-MYC by E6AP-dependent degradation of the repressor NFX1-91 (Figure 3) [140,141,142]. Epigenetic modulation of the *TERT* promoter is also thought to play a role in *TERT* induction; hypomethylation of the *TERT* transcription start site is a common biomarker of cervical cancer, and has been shown to have a functional impact on *TERT* gene expression [138,143,144,145]. Interestingly, the degree of *TERT* hypomethylation is dependent on the HPV type, with types belonging to clade A7 (HPV18-containing) showing little to no methylation, and types belonging to clade A9 (HPV16-containing) showing partial methylation [146]. The HPV oncoproteins have several epigenetic functional partners that may assist in *TERT* dysregulation, such as the DNA methyltransferase DNMT1, the histone acetyltransferase p300, and several histone deacetylases (HDACs), among many others [147,148].

## 4. Indirect Effects of Viral Infection on Genome Instability

### Genome Mutagenesis by APOBEC Enzymes

Genome-wide SNV patterns have been categorized into mutation signatures, many of which can be related back to a particular causative enzyme, replication error, and/or environmental mutagen [16]. Other than age-related mutations, the most prominent single nucleotide mutation signature in cervical cancers and other HPV-driven cancers is from the apolipoprotein B mRNA editing enzyme, catalytic polypeptide-like (APOBEC) enzyme family, a group of proteins used as a host defence mechanism in response to viral infection and endogenous retroviral activation [6,149]. APOBEC-family enzymes induce a characteristic cytosine deamination, which is seen widely across HPV-driven cancers including cervical cancer, HNSCCs, anal cancer, penile cancer and vulvar cancer, and can account for up to 68% of mutations within affected tumours [149,150,151,152,153,154]. APOBEC enzymes deaminate cytosine bases to uridine at single stranded DNA and RNA, which subsequently results in a C > T transition when inflicted on the genome [155]. Since the enzymes can only target single-stranded nucleotides, this mutagenesis usually occurs during DNA replication or at transcriptional bubbles; consequently, highly expressed genic regions are more prevalently targeted [156]. The APOBEC enzymes target the TC motif or a more specific TCW motif, with W representing either A or T [149]. Some APOBEC enzymes have gene-specific targets for their activity; for example, activation-induced cytosine deaminase (AID) is the enzyme responsible for somatic hypermutation and class switch recombination of the immunoglobulin genes on activated lymphocytes [157]. Other APOBEC enzymes, namely the APOBEC3 members, can induce mutations randomly across the genome; these are the enzymes responsible for APOBEC mutagenesis in most non-lymphocytic cancers [157]. In normal cells, APOBEC3 proteins (APOBEC3A/B/C/G/H) are important for host defence against exogenous retroviruses and DNA viruses (such as HPV), endogenous retroviruses, and active transposons [158,159]. A characteristic feature of APOBEC3 proteins is their tendency to alter the genome in strand-specific clusters, a phenomenon termed kataegis [160].

Similarly to their function in the normal human genome, APOBEC3 proteins’ antiviral activity as part of the host innate viral defence response also relies on their ability to induce viral hypermutation [161,162,163,164,165]. In fact, HPV genomes have evolved to limit the number of CT dinucleotides in an attempt to avoid damage by host APOBEC3 proteins [166]. Analysis of intra-host HPV sequence variability, i.e., the extent of sequence alteration of the HPV genome within a patient, has shown that HPV18 genomes are minimally affected by APOBEC3 mutagenesis while HPV16 genomes contain more APOBEC3-inflicted mutations, possibly indicating that HPV18 has evolved to better avoid APOBEC3 mutagenesis [167]. Introducing the HPV oncoproteins E6/E7 into keratinocyte cells can lead to increased *APOBEC3A*, *APOBEC3B*, and *APOBEC3H* mRNA expression [168], and as p53 levels reduce due to E6-dependant degradation, *APOBEC3B* levels increase [169]. Across cancers, the upregulation of *APOBEC3A* and *APOBEC3B* mRNA correlates with the level of APOBEC-inflicted mutations [149]. However, APOBEC3A protein levels are decreased in cervical tumour samples compared to precancerous samples, indicating that the antiviral effects of APOBEC3 proteins are restricted during tumorigenesis through a mechanism still unknown [170]. Although the APOBEC3 enzymes may lose their expression in cervical tumours, the damage they inflict remains in the tumour genome [151]. Cervical tumours also have strikingly different copy number profiles, in comparisons between tumours exhibiting predominant *CpG mutations versus the APOBEC3-mediated Tp*C mutations, which suggests that APOBEC3-mutagenesis may influence structural variants as well as point mutations [171]. HPV-positive HNSCCs have a significant increase in mRNA and protein expression of APOBEC3 members (3A, 3F, 3G, 3H) compared to their HPV-negative counterparts, and the expression of these genes can be even further increased upon interferon stimulation [172]. This differs from APOBEC3 expression in cervical cancer because it appears that the expression of APOBEC3 members are maintained within the established tumour, though HNSCCs do not have precancerous lesions that have been recovered and analyzed to compare to. APOBEC-mediated mutagenesis was found to be responsible for the PIK3CA-activating mutations E542K and E545K in HPV-positive HNSCC [173]. APOBEC3 activity, whether stimulated during viral infection or maintained within the established tumour, appears likely to play an important role in the mutagenesis of HPV-driven cancers.

## 5. HPV-Independent Mutations That Cause Genome Instability

### 5.1. Mutation of DNA Repair Genes

HPV-driven cancers have recurrent mutations in DDR genes that may further promote tumour genome instability. Recently, Halle et al. investigated genetic alterations in cervical cancers and found a particularly high frequency of mutations in DDR proteins [171]. The most frequently mutated DDR pathways were homology directed repair, the Fanconi anaemia pathway, nucleotide excision repair, and non-homologous end joining, with the most frequently mutated genes within these pathways including *TP53BP1* (4.0%), *POLQ* (4.0%), *BRCA2* (4.0%), *BRCA1* (3.7%), and *PRKDC* (7.4%) [171]. Germline small nucleotide polymorphisms in the Fanconi anaemia gene, *FANCA*, were also found to increase a person’s risk of developing CIN3 or cervical cancer by approximately 1.3–1.7 fold [174]. Mutations in DDR genes have also been reported in HNSCCs and are associated with advanced disease and a poorer prognosis [175,176]. HPV-positive HNSCCs more commonly harbored mutations in DNA repair mediators than HPV-negative HNSCCs, including in the genes *BRCA1*, *BRCA2*, *ATM*, and *FANCA* [177]. Many of these genes have been shown to be important for the episomal replication stage in the HPV life cycle, as discussed in Section 3.1. The increased prevalence of DDR gene mutations in later-stage disease may indicate that these somatic mutations are acquired after the tumour is established. After HPV is integrated into cellular DNA, it no longer requires the DDR machinery to replicate since it is replicated alongside chromosomal DNA. Thus, deregulation of these pathways may promote tumour evolution by increasing tumour genome instability once it is no longer dependent on the DDR for viral genome amplification.

### 5.2. Activating TERT Mutations

Activation of *TERT* expression in cancer can be induced through non-coding point mutations in the promoter sequence, leading to telomere lengthening [133]. Several cohorts of HPV-driven cancers, namely cervical cancer [113,178,179,180], head and neck cancers [180,181], vulvovaginal cancer [182], and penile cancer [183] have been reported to harbor hotspot somatic mutations in the *TERT* promoter. *TERT* promoter mutations occurred more frequently in the HPV-negative samples in cancer cohorts that included such cases (i.e., excluding cervical cancer). This is unsurprising since the genetic activation of *TERT* in an HPV-driven tumour would often be a redundant event (see Section 3.4). However, 95.7% of HPV-positive cervical cancers from an Indian cohort that had poor and moderately differentiated histopathology harbored a *TERT* promoter mutation, and such mutations also occurred at a higher frequency in cervical tumours from advanced-stage patients [180]. The genetic activation of *TERT* thus appears to more commonly allow HPV-negative cancer cells to achieve immortality in an HPV-independent context, but nevertheless also occurs in aggressive HPV-positive tumours.

## 6. Genomic of the Integrated HPV Genome

### 6.1. Mechanisms of HPV Integration

HPV integration into the host genome occurs as a consequence of increased genome instability in HPV-infected cells. Upregulation of *E6* and *E7* through E2 binding site methylation is thought to precede HPV integration [184,185]. These DNA methylation changes in the viral genome may initiate changes in gene expression that are required to generate genomic instability, which in turn increases the chances of HPV integration [186]. The frequency of integration increases during malignant progression and also varies by HPV type [187]. For instance, nearly all HPV18-infected tumours have evidence of HPV integration, while only 50–70% of HPV16-infected tumours display viral integration, indicating that oncogenic transformation of HPV18-infected cells but not HPV16-infected cells requires the viral genome to be integrated [20]. All high-risk HPV types have shown evidence of integration in cancers at varying frequencies [187]. Conversely, low-risk HPV types causative of genital warts (e.g., HPV9 and HPV11) rarely have detected viral integration except in cases of HPV-associated laryngeal papillomatosis, a rare and benign tumour affecting the upper respiratory tract of children [188,189,190]. Though not required for cancer, high rates of HPV integration is a defining attribute of high-risk HPV types and an important step in HPV-associated tumorigenesis.

HPV integration is generally regarded as an early and clonal event in the evolution of an HPV-driven tumour. In order for HPV to integrate, the virus must acquire double-stranded breaks to linearize its circular genome. One HPV copy may integrate into the host genome at a single location, or HPV genome segments may be distributed or duplicated across multiple locations in the host genome; the latter are referred to as integration events. The most common viral genome breakpoints found in tumours are within the *E2* or *E1* genes [43]. Conversely the *E6* and *E7* genes are almost never interrupted by breakpoints. E2, which is the main negative regulator of viral oncogene expression [21,44], is thus commonly inactivated by these breakpoints. The viral long control region, which precedes the early promoter and which is rich in transcription factor motifs that contribute to activating transcription, is also frequently maintained [44]. Dysregulation of human genes via HPV integration may also confer cancer-promoting properties; indeed, activating HPV integrations near cervical cancer oncogenes (e.g., *TP63* and *MYC*) and tumour suppressors (e.g., *RAD51*) have been observed, indicating that integration can both amplify and disrupt human gene expression [21,43,191,192]. HPV integration tends to be observed in transcriptionally active regions of open chromatin rather than inactive heterochromatic regions, as well as at or near fragile sites [192]. Integrations within repeat regions such as telomeres and centromeres have also been observed, but these are more difficult to characterise using short-read sequencing technology [193].

The mechanism(s) by which the HPV genome is integrated into the host genome and subsequently rearranged is unclear, but two models have emerged in the last few years. The “looping” model postulates that HPV integrates to form a bridge between non-contiguous human DNA at two sites with double-stranded breaks [19] (Figure 4a). The region containing HPV then forms a transient loop structure that is amplified during DNA replication [19,194]. Almost all HPV integrations seem to be associated with two double stranded human breakpoints, between which the human DNA can be either deleted or amplified. The second model proposes that HPV hijacks DDR mechanisms to facilitate integration at regions of microhomology that acquire double-stranded breaks (Figure 4b) [43]. Hu et al. found that human genomic regions that share ≥four base pairs of microhomology with the HPV sequence had a significant enrichment of HPV breakpoints [43]. They hypothesized that this meant that viral integration was the result of cervical cells being reliant on error-prone DNA repair pathways such as fork stalling and template switching and microhomology-mediated break-induced replication, which caused accidental integration when repairing double-stranded DNA breaks [43]. The microhomology-mediated end-joining pathway is an error-prone method for mending double stranded breaks and often is only used as a backup if HR is not able to repair a break [195]. As discussed previously (Section 3.1), HPV genomes use the HR machinery in order to replicate and often sequester the repair machinery away from human genome breaks. HR, although relied upon for episomal replication, has not yet been implicated in HPV integration. However, the process by which the DDR pathway switches from high-fidelity homology-directed repair to more error-prone methods that result in viral integration has not been elucidated. Other factors can also affect the likelihood of HPV integration. Increased levels of oxidative stress, either through E6-dependent induction or nutrient depletion in the media, leads to an increased frequency of foreign DNA integration [196,197]. Additionally, APOBEC3 expression increases the likelihood of HPV integration occurring in HNSCCs *in vitro*, though the mechanism by which this occurs is not known [172]. HPV-infected cells require double-stranded breaks in both the human and viral genomes for accidental viral integration to occur; therefore, it is plausible that additional sources of DNA damage aid the integration process.

### 6.2. Genome Instability at Regions of HPV Integration

The whole-genome sequencing of HPV-integrated cancer cells has shown that structural alterations, including duplications, deletions, translocations, and inversions, are often flanked or bridged by HPV integrants (Figure 4c) [19,21]. The entire loci surrounding HPV integration are often amplified on the integrated haplotype, though other copy number changes such as loss of heterozygosity (LOH) have also been reported [198,199]. Haplotype-specific amplifications result in the high expression of human genes near the integrated viral genome, thus potentially upregulating human oncogene expression [198]. Some samples contain multiple independent HPV integration events at different locations across the tumour genome, which could hypothetically act as *de novo* regions of homology that could promote non-allelic homologous recombination errors. During the integration process, a human-HPV bridge may be excised and exist as self-replicating extrachromosomal circular DNA (ecDNA). Evidence of ecDNAs has been found in many other cancer types and involves a genome fragment, often containing a small oncogene such as *MYC*, which is excised and amplified [200,201,202]. In the case of HPV, two or more double-stranded breaks excise a segment of the human genome, which is then bridged with the linearized HPV genome, thus creating a hybrid human-HPV circular structure (Figure 4c) [202]. In principle, the episome could incorporate human cis regulatory elements, which might help drive viral oncogene expression.

The HPV genome may integrate as a single full copy, a portion of the genome, or as multiple repeating or rearranged segments. Multiple neighbouring integration sites may exist as a single locus. Kadaja et al. first showed that E1 and E2 drive a re-replication of the HPV genome that leaves a heterologous population of intermediate integrant structures [203]. This “onion-skin” type replication can leave the DNA in a linear, supercoiled, or open circular structure (Figure 4c) [203]. The intermediates must then be resolved by DDR machinery, which can be seen coating the E1-replicated integrant structures (both HR and non-homologous end joining), and which often results in structural rearrangements once resolved [203]. A recent study used Oxford Nanopore Technology long-read sequencing to show HPV episomes that exist in multimer form; they consisted of repeating and/or rearranged segments of the HPV genome concatenated together in extrachromosomal DNA [204]. The authors hypothesized that the HPV multimers form as an intermediate prior to integrating into the human genome. This study provides evidence that HPV genome amplification and rearrangement could, at least in some cases, occur before it is integrated into the human genome [204]. Overall, integrated HPV genomes have been shown to accumulate significant rearrangements before, during, and after the act of integration.

## 7. Targeting Genome Instability in HPV-Driven Cancer Treatment

Genome instability may drive tumour evolution, but too much genome instability can lead to cell death. This double-edged sword can be exploited in cancer therapy by driving mechanisms that induce genome instability in tumours that are unable to repair the damage (such as HR deficiency). For instance, the DDR is important for both HPV infection and carcinogenesis, and has been targeted by therapy to treat both contexts. HPV infection was modelled in HPV18 organotypic raft cultures, and inhibition of the ATR/Chk1 and ATM/Chk2 pathways was shown to inhibit re-entry into the cell cycle and reduce viral amplification by 90–99% [205]. The inhibition of DDR proteins in cervical cancer and HPV-positive HNSCC has repeatedly been found to sensitize cells and tumours to therapy. Within cervical cancer cells and xenografts, inhibition of the protein encoded by the HR mediator gene *RAD51* (which is recurrently targeted by HPV integration) can reduce tumour growth, slow cell cycle progression, and sensitize the tumours to radiation and cisplatin [206]. Targeting the base excision repair pathways was found to be effective in sensitizing HPV-positive but not HPV-negative HNSCCs to radiation, while the inhibition of nonhomologous end-joining and mismatch repair was effective for both [207]. Alsahafi et al. found that HPV-positive HNSCC cell lines and tumour models with increased sensitivity to radiotherapy had an overexpression of EGFR that correlated with down regulation of E6 and the DDR [208]. Another study by Liu et al. attributed increased therapeutic response to radiation and chemotherapeutics in HPV-positive HNSCCs to a loss of TGF-β signalling [114]. The authors found that the TGF-β pathway repressed the microRNA miR-182, which targets the HR effector genes *BRCA1* and *FOXO3*; therefore, loss or inhibition of TGF-β signalling led to the inhibition of HR and increased tumors’ reliance on other, error-prone repair methods [114]. A commonly targeted DDR enzyme in HR-deficient tumours is PARP, which functions in the “backup” non-homologous end-joining pathway [209]. In HNSCCs, HPV-negative tumour models have been found to have a similar if not better response to PARP inhibition than HPV-positive tumours; thus, PARP inhibitor sensitivity has not been concluded to be a consistent feature of HPV-positive cancers [210,211]. Another therapeutic target in tumours is the WEE1 kinase. WEE1 inhibits activity of both CDK1, which initiates the G2/M checkpoint, and CDK2, which mediates the S phase checkpoint after checking for DNA damage, thus making it an attractive oncotarget for tumours displaying enhanced or uncontrolled genome instability [212,213]. WEE1 kinase inhibitors have been used in combination to sensitize HNSCCs to other therapies such as radiotherapy, chemotherapy, and immunotherapy [210,214,215]. Recently, Diab et al. showed that the introduction of the *E6* and *E7* oncogenes into HNSCC cells also induces sensitivity to WEE1 inhibition [216]. The authors found that E6/E7 promoted an upregulation of the two cell cycle regulators FOXM1 and CDK1, which resulted in a dependance on WEE1-mediated regulation. Thus, the inhibition of WEE1 resulted in uncontrolled entry into mitosis and increased DNA damage that was detrimental to cell survival. Targeting the increase in genome instability in HPV-driven cancers through modulation of the DDR pathways as a means to increase sensitivity to radiation and chemotherapy may be a promising avenue of cancer treatment.

## 8. Conclusions

Genome instability is one of the central hallmarks of cancer; however, the mechanism by which cells generate genome instability varies between tumours and cancer types. HPV-driven cancers have the element of a ubiquitous exogenous virus that creates a specific mutagenic environment. The HPV genes are able to dysregulate human pathways that are important for maintaining genome integrity; eventually, this leads to further genetic mutations and the onset of cancer. A consequence of the increase in genome instability is the integration of the HPV genome into the host genome. Once integrated, the HPV genome and the region around it is prone to significant rearrangements. Targeting the pathways that are commonly utilized for generating genome instability in HPV-driven cancers may be a novel therapeutic avenue in this context. All together, the mechanisms in which genome instability is initiated and maintained within HPV-driven cancers is multifaceted. Identifying the DNA damage within the HPV-driven tumours and understanding the mechanisms that lead to it will help in our understanding of how these cancers develop and progress.

## Figures and Tables

**Figure 1 cancers-14-04623-f001:**
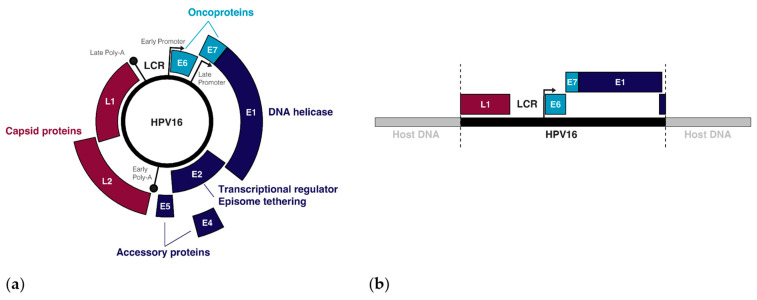
The episomal and integrated forms of the HPV genome: (**a**) The genes and regulatory regions in the HPV16 episome; (**b**) An example of the integrated form of HPV16 with breakpoints in the *L1* and *E2* genes.

**Figure 2 cancers-14-04623-f002:**
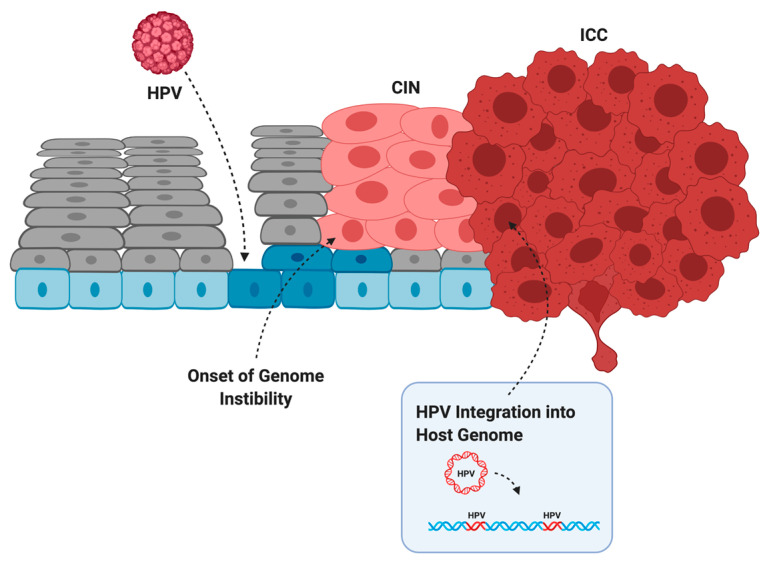
The evolution of HPV infection, to cervical intraepithelial neoplasia (CIN), to invasive cervical cancer (ICC). HPV infects basal cells (blue) through microabrasions in the cervical epithelium. Uncontrolled proliferation of the mid-epithelial layers allows the onset of genome instability. After acquiring additional somatic mutations and often HPV integration, the tumor breaks through the basal layer and is graded as an ICC. Created with BioRender.com.

**Figure 3 cancers-14-04623-f003:**
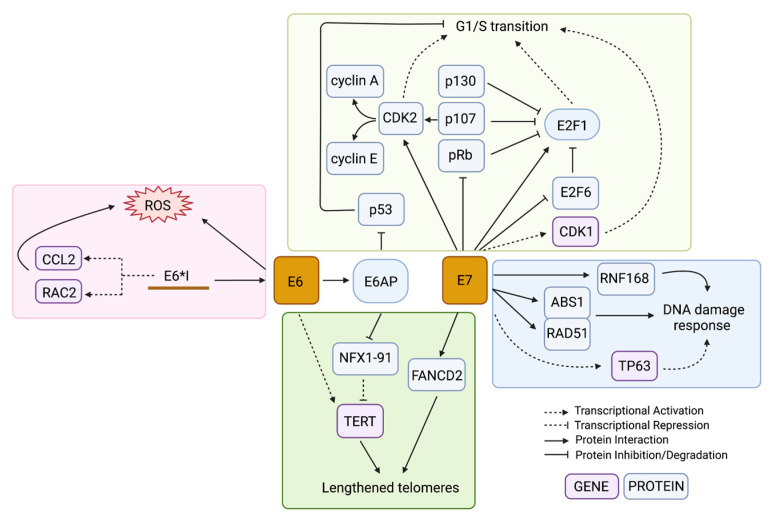
Mechanisms in which E6 and E7 lead to genome instability. The HPV oncoproteins E6 and E7 dysregulate several pathways at both the gene and protein level as a way to increase genome instability in the infected cell, including the cell cycle transition (yellow box), DDR pathways (blue box), generation of oxidative stress (red box), and telomere length alterations (green box). The colour of the box containing the gene name indicates if it is being regulated at the gene or protein level. Created with BioRender.com.

**Figure 4 cancers-14-04623-f004:**
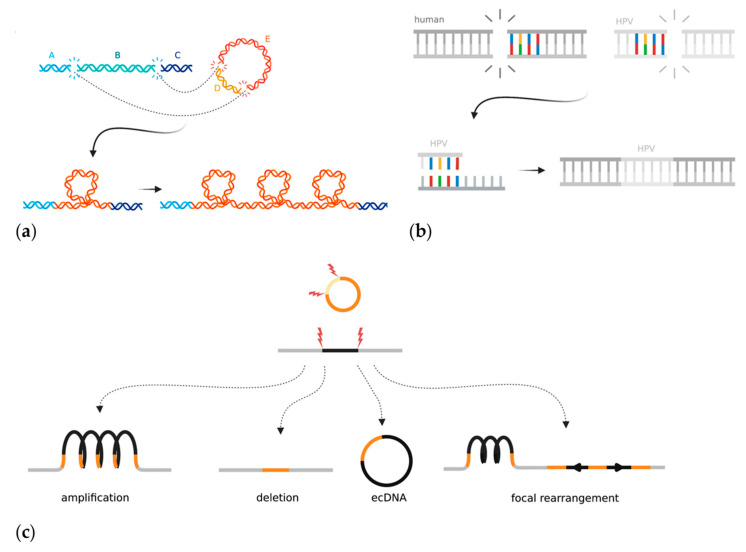
Mechanisms and local consequences of HPV integration in cancer: (**a**) The looping model proposes that HPV acts as a bridge between two pieces of non-contiguous DNA that have double-stranded DNA breaks in the human genome. The looped bridge can then be successively amplified upon DNA replication; (**b**) The microhomology model proposes that regions of microhomology between the human genome and HPV mediate integration through DNA repair mistakes in the fork stalling and template switching and microhomology-mediated break-induced replication pathways; (**c**) The integrated HPV genome can induce types of local rearrangements at the site of integration. Created with BioRender.com.

## Data Availability

Not Applicable.

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
