# Peer review of "The Drivers, Mechanisms, and Consequences of Genome Instability in HPV-Driven Cancers"

_cancers, 2022, doi:10.3390/cancers14194623_

Round 1
Reviewer 1 Report
In the manuscript “The drivers, mechanisms, and consequences of genome instability in HPV-driven cancers”, Porter VL and Marra MA review the HPV-dependent and HPV-independent mechanisms involved with the HPV-cancer induction. This is an important review that puts together important concepts and gives an actualized update on the field. The manuscript is well written, and all the topics are consistent with the proposed topic. I recommend the publication of this manuscript at this journal, however the only missing topic on this review is the oncogenes E6 and E7 splicing. Overexpression of these oncogenes are required for tumorigenesis, and the alternative splicing of the intron localized at E6 ORF, generating the E6*I RNA, is required for E7 expression. I would suggest to the authors include a section about the E6/E7 splicing, which will contribute to make this review paper complete.
Reviewer 2 Report
This is a well-written review of genomic instability in HPV-driven cancers. The review is nicely structured and provides good detail on the direct and indirect effects of HPV on different genomic instability mechanisms.
Comments:
Abstract: Please keep it generic- remove the reference to APOBEC3 enzymes and cytosine deamination. APOBEC3 is only one of the many things discussed in the review, hence mentioning this in the abstract is a bit of distraction.
The paper has no mention of E5 and its role as an oncogene. There is substantial evidence E5 can contribute to oncogenic transformation (PMID: 22078316), particularly in cervical cancer (PMID: 20332225, PMID: 23128977). This would be an important subject to touch on, particularly in section 3.1.
If data is available it would be interesting to highlight the importance and prevalence of each genomic instability mechanism in the different HPV-driven cancers, potentially displayed as a table.
There could be mention of the role of innate immunity in DNA damage responses in section 3.2. STING (PMID: 29622565; PMID: 33903123) and TLRs (PMID: 22931928) respond to DNA damage and induce anti-tumour inflammatory responses. There is evidence HPV targets these immune sensors (PMID: 34372596, PMID: 33903123, PMID: 29263932) which may aid in the dysregulation of the DNA damage response.
Figures are appropriate, but figure 3 could have more detail in the figure legend.
Lines 536 and 543 refer to figure 3, but should be figure 4.
Reviewer 3 Report
The authors have reviewed several mechanisms of genome instability in Human Papillomavirus (HPV)-associated cancers. They discussed how these mechanisms drive the tumor at the whole-genome level and around sites of HPV integration.
The review topic is well-timed as the latest findings have come up in recent times complementing the previous research. This would be of immense help to the field.
Overall, the authors have done excellent work in composing the review with all the relevant data and appropriate figures.
Comments:
The authors may discuss the DNA methylation on the HPV genome as a potential contributor to HPV integration and consequently genomic instability across HPV tumors.
Reviewer 4 Report
In this manuscript, Porter et al. investigate the mechanisms and clinical outcomes of genomic instability in HPV-driven cancers. This is a comprehensive and informative review that will be a useful resource for the HPV community. I have only a few comments.
- In the abstract, the authors state 'The HPV genome often becomes integrated into the host genome during HPV-induced tumorigenesis...'. This is not necessarily a common occurance in some HPV cancers, such as head and neck cancer where HPV exists as an non-integrated episome in around 75% of cases (Nulton et al., Oncotarget, 2017). This should be discussed and the abstract clarified.
- Reference 2 - an updated version of this classic paper should also be referenced here (Hanahan, Cancer Discovery, 2022)
- Line 63-66 - again, studies have shown that in head and neck cancer, many HPV positive cases do not contain integrated HPV genomes
- 2. A recent review on HPV-mediated transformation would be useful here (Scarth et al., J Gen Virol, 2021)
- The WRN protein has recently been implicated in the HPV life cycle (Das et al., mBio, 2019' James et al., mSphere, 2020). As an important regulate of the DDR, these studies should be referenced.
- The WEE1 G2/M checkpoint kinase is up-regulated in HPV+ HNSCC (Diab et al., PNAS, 2021; Zu et al., Mol Cancer Res, 2022). High WEE1 sensitises these tumors to WEE1 inhibition and this is particularly effect in tumors with high chromosomal instability - these studies should be discussed as potential therapeutic approaches.
